# Deciphering Molecular and Phenotypic Changes Associated with Early Autoimmune Disease in the Aire-Deficient Mouse Model of Sjögren’s Syndrome

**DOI:** 10.3390/ijms19113628

**Published:** 2018-11-17

**Authors:** Feeling YuTing Chen, Eliza Gaylord, Nancy McNamara, Sarah Knox

**Affiliations:** 1Department of Cell & Tissue Biology, University of California San Francisco, San Francisco, CA 94143, USA; feeling.chen@ucsf.edu (F.Y.C.); eliza.gaylord@ucsf.edu (E.G.); 2School of Optometry and Vision Science Graduate Program, University of California, Berkeley, CA 94720, USA; 3Department of Anatomy, University of California San Francisco, San Francisco, CA 94143, USA

**Keywords:** Sjögren’s syndrome, disease progression, pathway activation, Aire mouse

## Abstract

Sjögren’s syndrome (SS) is characterized by extensive lymphocytic infiltration of the salivary and lacrimal gland (LG), resulting in acinar cell destruction and organ dysfunction. The underlying pathogenesis of SS remains largely unknown, and studies historically focus on defining late-stage disease. Here, we identify tissue programs associated with disease onset using transcriptomic and immunohistological analysis of LGs from 5- and 7-week-old mice deficient in autoimmune response element (Aire). At 5 weeks of age (wk), Aire-/- mice show minimal tissue dysfunction and destruction compared to 7 wk Aire-/-, which exhibit severe dry eye, poor tear secretion, extensive lymphocytic infiltration, reduced functional innervation, and increased vascularization. Despite this mild phenotype, 5 wk Aire-/- LGs were highly enriched for signaling pathways previously associated with SS, including interferon gamma (IFNγ), interleukin 1 beta (IL1β), nuclear factor kappa-light-chain-enhancer of activated B cells (NF-κB), toll-like receptor (TLR) signaling, and interleukin-6/signal transducer and activator of transcription 3 (IL6/STAT3) signaling. Novel signaling pathways such as the semaphorin–plexin pathway were also noted. Intriguingly, we found an expansion of the ductal network with increasing disease. Activated STAT3, a blocker of apoptosis, was restricted to the ductal system and also increased with damage, highlighting its potential as a promoter of ductal cell survival. These data demonstrate the early activation of signaling pathways regulating inflammation, innervation, and cell survival before the onset of clinical disease indicators, suggesting their potential value as diagnostic biomarkers.

## 1. Introduction

Sjögren’s syndrome (SS) is a complex autoimmune exocrinopathy that has been extensively reported in the clinical literature. Although a systemic disorder, it predominately affects the salivary and lacrimal glands (LG) with lymphocytic infiltration of these tissues resulting in clinical manifestations that include xerostomia (dry mouth) and keratoconjunctivitis sicca (dry eyes) [1]. While several important discoveries have been made in the clinical diagnosis and management of SS, many of the molecular pathways that mediate the pathogenesis of SS-associated exocrinopathy remain poorly articulated. This is due, in part, to a limited understanding of the genetic framework that underpins the disease process and the lack of meaningful genetic biomarkers that are strongly associated with the disease. Equally problematic is the lack of progress in defining early markers of disease, with diagnosis being largely dependent on late-stage indicators such as blood testing for anti-Sjögren’s syndrome-related antigen A (SSA/Ro), Schirmer test for lacrimal function, and measurements of saliva flow. In many cases, analysis of labial salivary gland biopsies takes place well after the disease has been initiated and extra-glandular organ involvement has already occurred, thereby limiting treatment options for SS patients.

Although some progress has been made in the phenotypic and genomic characterization of SS, including a recent genome-wide association study [2,3], causative factors remain elusive. Analyses of patient biopsies and mouse models of SS have identified a large number of immune signaling pathways that may underlie the disease, such as activation of IFNγ producing T-helper 1 (Th-1) [4,5] and IL17 producing T-helper 17 (Th-17) CD4+ T cells [6,7], B cell activation [8,9], upregulation of pro-inflammatory cytokines (e.g., IL1, IL6, and IL17) [6,7,10,11], ligation of Toll-like receptors expressed on infiltrating immune and ductal epithelial cells [12,13], phosphorylation of signal transducer and activator of transcriptions (STATs) [13], and signaling via the NF-κB pathway [13]. Also notable is the discovery of a link between the nervous and immune system mediated by neuroimmune semaphorins (e.g., Sema4A, 4D, 6D, and 7A). While originally identified for their signaling role in axon guidance through direct binding of plexin receptors, more recently, semaphorins have been shown to provide finetuning of inflammatory processes that are altered in autoimmune diseases, making them interesting molecules for immunotherapy [14,15,16].

To gain a better understanding of global gene expression and how it is altered in the early stages of SS exocrinopathy, we used a well-characterized, spontaneous mouse model of SS and state-of-the-art genetic screening methods to conduct an unbiased exploration of global gene expression. In comparison to the majority of SS mouse models, mice deficient in autoimmune regulator gene (Aire) exhibit classic signs of aqueous-deficient dry eye in a swift fashion, allowing for an understanding of disease processes in the absence of the effects of aging. By 7–8 weeks the lacrimal glands of Aire-/- mice exhibit extensive lymphocytic infiltration combined with corneal pathologies and severe dry eye. However, it is not known when these disease outcomes are initiated in the Aire-/- mouse. Aire-/- mice on the BALB/c background were selected over Aire-/- mice on the NOD background for this study due to their less rapid development of severe disease. NOD. Aire-/- mice develop multi-organ autoimmune disease that results in an average life span of ~8 weeks. In contrast, BALB/c. Aire-/- mice develop SS-associated dry eye and exocrinopathy that is nearly identical in appearance to NOD. Aire-/-, but the time course of disease development is slower, the extent of multi-organ involvement is less, and the average life span extends up to ~16 weeks. These characteristics make BALB/c. Aire-/- mice an ideal model for studies of disease progression.

Here, we used RNA sequencing (RNAseq), the gold standard for generating quantitative estimates of RNA abundance and disease-specific alterations in gene expression, to examine the entire transcriptome of CD4+-T-cell-mediated exocrinopathy in 5-week-old (wk) Aire-/- mice. While dry eye researchers have used RNAseq technology to explore the transcriptome of ocular tissues, none have focused on the specific alterations in SS-associated exocrinopathy during the early stages of disease or gone on to validate differences in gene expression using traditional readouts, such as qPCR or protein expression (e.g., immunohistochemistry or Western blot). Our findings provide a holistic overview of genetic changes associated with early disease processes, as well as specific genetic biomarkers that can be used to define the mechanism of disease development and progression.

## 2. Results

### 2.1. Aire-/- at 5 Weeks of Age Exhibit Mild Lacrimal Gland Disease Compared to Mice at 7 Weeks 

In order to begin to define pathways activated during early disease onset we utilized the Aire-/- mice which exhibit epithelial barrier disruption, poor tear secretion, extensive lymphocytic infiltration, and vascularization of both lacrimal glands and cornea, as well as severe corneal pathologies and loss of lacrimal acini by 8 weeks of age (wk) as compared to age-matched wild types (WT) (Figure 1) [17]. To determine when these outcomes are initiated, we measured lissamine green staining (indicator of epithelial barrier function) and stimulated tear production in Aire-/- and WT mice at 5 and 7 wk. Although lissamine green staining was increased in both the 5 and 7 wk Aire-/- cornea, this difference was greatly pronounced in 7 wk tissue, indicating extensive damage by this time point (5 wk: 1.00 ± 0.15 vs. 2.71 ± 0.26, *p* = 0.0013; 7 wk: 1.00 ± 0.37 vs. 8.13 ± 0.30, *p* < 0.0001, Figure 1A). Consistent with the lissamine green outcomes, stimulated tear production was markedly decreased in 7 wk Aire-/-, whereas tear levels were similar to WT in 5 wk Aire-/- animals (5 wk: 1.00 ± 0.094 vs. 0.75 ± 0.19, *p* = 0.34; 7 wk: 1.00 ± 0.055 vs. 0.20 ± 0.065, *p* < 0.001, Figure 1A, right). As these results suggested mild disease at 5 wk and severe disease at 7 wk, we then characterized inflammation of the 5 and 7 wk Aire-/- lacrimal gland and cornea. As expected, we measured extensive CD4+ T cell inflammation of the cornea in 7 wk Aire-/- tissue, with CD4+ T cells infiltrating the cornea (0.0 ± 0.0 vs. 8.66 ± 2.99, *p* = 0.03, Figure 1B), while both CD4+ T cells (0.0 ± 0.0 vs. 25.60% ± 5.04, *p* = 0.002, Figure 1C) and CD45R+ B cells densely infiltrated throughout the lacrimal gland (0.0 ± 0.0 vs. 30.05% ± 6.02, *p* = 0.004, Figure 1D). In contrast, at 5 wk there were few to no immune cells in the cornea (0.0 ± 0.0 vs. 1.17 ± 0.83, *p* = 0.22, Figure 1B) and fewer and more restricted foci of T (0.0 ± 0.0 vs. 6.42% ± 0.58, *p* < 0.0001, Figure 1C) and B cells (0.0 ± 0.0 vs. 6.11% ± 1.21, *p* = 0.015, Figure 1D) in 5 wk Aire-/- lacrimal glands. We previously found an increased number of dilated blood vessels within the intact epithelial region during disease progression (5 wk: 0.50 ± 0.50 vs. 3.40 ± 1.21, *p* = 0.07; 7 wk: 1.00 ± 0.71 vs. 2.00 ± 0.55, *p* = 0.31 Figure 1E), consistent with chronic inflammation; however, there was no statistically significant difference in vessel diameter between 5 and 7 wk Aire-/- lacrimal glands compared to age-matched wild-type mice (3.40 ± 1.21 vs. 2.00 ± 0.55, *p* = 0.33, Figure 1E, right).

In order to begin to define pathways activated during early disease onset we utilized the Aire-/- mice which exhibit epithelial barrier disruption, poor tear secretion, extensive lymphocytic infiltration, and vascularization of both lacrimal glands and cornea, as well as severe corneal pathologies and loss of lacrimal acini by 8 weeks of age (wk) as compared to age-matched wild types (WT) (Figure 1) [17]. To determine when these outcomes are initiated, we measured lissamine green staining (indicator of epithelial barrier function) and stimulated tear production in Aire-/- and WT mice at 5 and 7 wk. Although lissamine green staining was increased in both the 5 and 7 wk Aire-/- cornea, this difference was greatly pronounced in 7 wk tissue, indicating extensive damage by this time point (5 wk: 1.00 ± 0.15 vs. 2.71 ± 0.26, *p* = 0.0013; 7 wk: 1.00 ± 0.37 vs. 8.13 ± 0.30, *p* < 0.0001, Figure 1A). Consistent with the lissamine green outcomes, stimulated tear production was markedly decreased in 7 wk Aire-/-, whereas tear levels were similar to WT in 5 wk Aire-/- animals (5 wk: 1.00 ± 0.094 vs. 0.75 ± 0.19, *p* = 0.34; 7 wk: 1.00 ± 0.055 vs. 0.20 ± 0.065, *p* < 0.001, Figure 1A, right). As these results suggested mild disease at 5 wk and severe disease at 7 wk, we then characterized inflammation of the 5 and 7 wk Aire-/- lacrimal gland and cornea. As expected, we measured extensive CD4+ T cell inflammation of the cornea in 7 wk Aire-/- tissue, with CD4+ T cells infiltrating the cornea (0.0 ± 0.0 vs. 8.66 ± 2.99, *p* = 0.03, Figure 1B), while both CD4+ T cells (0.0 ± 0.0 vs. 25.60% ± 5.04, *p* = 0.002, Figure 1C) and CD45R+ B cells densely infiltrated throughout the lacrimal gland (0.0 ± 0.0 vs. 30.05% ± 6.02, *p* = 0.004, Figure 1D). In contrast, at 5 wk there were few to no immune cells in the cornea (0.0 ± 0.0 vs. 1.17 ± 0.83, *p* = 0.22, Figure 1B) and fewer and more restricted foci of T (0.0 ± 0.0 vs. 6.42% ± 0.58, *p* < 0.0001, Figure 1C) and B cells (0.0 ± 0.0 vs. 6.11% ± 1.21, *p* = 0.015, Figure 1D) in 5 wk Aire-/- lacrimal glands. We previously found an increased number of dilated blood vessels within the intact epithelial region during disease progression (5 wk: 0.50 ± 0.50 vs. 3.40 ± 1.21, *p* = 0.07; 7 wk: 1.00 ± 0.71 vs. 2.00 ± 0.55, *p* = 0.31 Figure 1E), consistent with chronic inflammation; however, there was no statistically significant difference in vessel diameter between 5 and 7 wk Aire-/- lacrimal glands compared to age-matched wild-type mice (3.40 ± 1.21 vs. 2.00 ± 0.55, *p* = 0.33, Figure 1E, right).

### 2.2. Gene Expression Analysis Reveals Multiple Signaling Pathways Upregulated in the Lacrimal Glands of Mice Exhibiting Mild Disease

Current studies have pointed to a number of specific pathways being active in late stage disease, yet whether these are also present at early stages is unclear. To address this issue, we compared gene expression in lacrimal glands from 5 wk Aire-/- with very mild disease, i.e., those with lymphocytic infiltrates ≤10% (Figure 1C) to their age-matched wild-type controls via bulk RNA sequencing (Appendix A). As shown in Figure 2A, principal component analysis of LG samples revealed transcriptomes clustered as discrete groups based on their genotype, with some variability within groups. Our differential expression analysis demonstrated extensive changes in gene expression, with an overall increase in global expression as shown by 1909 genes being upregulated (>2-fold; *p* < 0.01) compared to 694 being downregulated (<−2; *p* < 0.01; Figure 2B). Not surprisingly, gene ontology analysis indicated a significant enrichment in genes associated with the adaptive and innate immune responses (Figure 2C, Table A1). These include genes involved in T and B cell activation and proliferation, chemotaxis, antigen presentation and processing (MHC1), macrophage activation and phagocytosis, regulation of T-helper cells (type 1), mast cell and neutrophil activation, and cytokine production and signaling (for a list of representative genes, see Table 1; for complete list of genes, see Appendix A). In line with our immunostaining for inflammation and blood vessels, we identified upregulation of gene sets involved in hematopoiesis and lymphoid organ development, indicating the creation of germinal centers as well as vascularization. Interestingly, we also found an enrichment for actin cytoskeleton genes, suggesting that the tissue was remodeling. Our unbiased analysis also revealed a significant enrichment of numerous inflammatory signaling pathways that have been reported to occur in the advanced-stage human disease as well as in a number of SS mouse models (including the Aire-/-). These include IFNγ [18,19], NF-κB [13], toll-like receptor signaling [12], interleukin (e.g., IL1β, IL2, IL6), and JAK/STAT [13] (see Table 1 for representative genes). In addition to these pathways, we also found a small but significant downregulation of secretory genes including *Pip* (Fold change (FC) = −2.3, *p* = 0.00032), secretoglobulins (e.g., *Scgb1b3*, FC = −3.57, *p* = 0.0002), and exocrine gland secreted peptides (e.g., *Esp6*, FC = −2.9, *p* = 0.0058), as well as the acinar marker and master regulator of secretion Basic Helix-Loop-Helix Family Member A15 (*Bhlha15/Mist1*; FC = −2.04, *p* = 0.0078); this suggests that, despite tear secretion being similar to in wild-type mice, LG function was being reduced. Concurrent with the loss of acinar genes was an upregulation of the ductal markers mast/stem cell growth factor receptor (*Kit*; FC = 2.5, *p* < 0.00005) and solute carrier 12 member 4 (*Slc12a4*/KCC1; FC = 2.2, *p* < 0.005). These results suggest that remodeling of the secretory tissue and alterations in the proportion of acinar and duct cells in LG begins at or before 5 weeks.

### 2.3. Validation of Gene Expression Data Confirms the Presence of Multiple Inflammatory Pathways Apparent at Early Stages of Disease Development

To validate the alterations in gene expression/pathways/processes measured via transcriptomic analysis, as well as to define their spatiotemporal distribution and abundance with increasing disease, we performed a combination of Western blot and immunofluorescence in 5 and 7 wk Aire-/- and WT LG. Immunofluorescent analysis confirmed the reduction in the acinar cell compartment and acquisition of a more ductal phenotype during disease progression (Figure 3A, top and middle panels), with the accumulation of KIT+ sodium potassium calcium co-transporter 1 (NKCC1)+ ductal tissue in the Aire-/- (Figure 3A) and the reduction in MIST1+ acini (5 wk: 4.63 ± 0.69 vs. 1.44 ± 0.14, *p* = 0.02; 7 wk: 4.64 ± 0.86 vs. 0.70 ± 0.09, *p* = 0.02, Figure 3B). Surprisingly, we found MIST1 to be co-expressed in NKCC1+ cells in Aire-/- LG at 7 wk (Figure 3B, arrowheads). Closer analysis at 5 wk showed only a few of these co-expressing cells, suggesting either that ductal cells are transdifferentiating towards the acinar lineage or that acinar cells begin to express ductal markers. To determine if the proportion of duct cells was increasing via proliferation or via loss of acini, we immunostained for the cell proliferation marker Ki67 and the apoptotic marker cleaved caspase-3 (CC3). We found a significant increase in duct cell proliferation (5 wk: 0.98% ± 0.60 vs. 6.12% ± 1.59, *p* = 0.04; 7 wk: 0.33% ± 0.33 vs. 9.90% ± 2.33, *p* = 0.02, Figure 3C) and limited cell death (5 wk: 0.11% ± 0.11 vs. 0.31% ± 0.14, *p* = 0.28; 7 wk: 0.19% ± 0.11 vs. 0.92% ± 0.16, *p* = 0.01, Figure 3D) in both the 5 and 7 wk LG compared to WT LG. In contrast to this finding, acinar cells showed little proliferation and extensive apoptosis at both ages (5 wk: 0.25% ± 0.14 vs. 4.50% ± 0.52, *p* < 0.001; 7 wk: 0.50% ± 0.20 vs. 4.41% ± 1.23, *p* = 0.03, Figure 3D), confirming our previous findings at 8 wk that the acini but not the ducts are adversely impacted by chronic inflammation in the Aire-/- mouse model.

Given the overrepresentation of the janus kinase/signal transducers and activators of transcription (JAK/STAT) pathways in the Aire-/- LG at 5 wk (Figure 2C) and the involvement of the STAT pathway in promoting cell survival, we next asked whether duct cell survival was due to increased phosphoSTAT3 (pSTAT3), a potent inhibitor of cell apoptosis. Unexpectedly, immunostaining of WT tissue for pSTAT3 showed extensive phosphorylated (active) protein within ductal cells but not in acinar cells. Furthermore, in the 5 wk Aire-/- LG we found increased numbers of pSTAT3+ immune cells (Figure 4A), as well as pSTAT3+ ductal cells (5.63 ± 1.80 vs. 35.75 ± 7.29, *p* = 0.02, Figure 4B). The number of pSTAT3+ ductal cells was similar at 5 and 7 wk (35.75 ± 7.29 vs. 46.6 ± 6.35, *p* = 0.3, Figure 4B), confirming the activation of the JAK/STAT signaling pathway early in the disease process and likely mediated by infiltrating immune cells. Although we also observed an increase in Il6 receptor (Il6ra) transcripts (FC = 7.40, *p* < 0.0001, Table 1) and IL6Ra protein at 5 wk (0.97 ± 0.03 vs. 6.97 ± 1.99, *p* < 0.05, Figure 4B), a well-known activator of STAT3, immunostaining revealed an enrichment of the receptor in immune cells but not in ductal cells, indicating that another factor was promoting the phosphorylation of STAT3 in these cells. In addition, *Stat1* was also significantly upregulated during early disease (FC = 12.07, *p* < 0.00005, Table 1) and in line with elevated STAT1 protein level by Western blot (5 wk: 0.94 ± 0.19 vs. 6.58 ± 1.10, *p* = 0.01; 7 wk: 0.89 ± 0.07 vs. 3.65 ± 0.55, *p* = 0.01, Figure 4C). STAT1 is known to be activated by signaling from INFs, and advanced studies on the development of JAK/STAT inhibitors have found that expression of the STAT1 pathway confers cellular resistance to DNA-damaging agents and supports tumor growth [20]. Together these data suggest that ducts may evade cell death through activation of the STAT3 pathway.

### 2.4. Denervation of the LG Begins at Early Disease Stages

The GO analysis revealed an increase in the semaphorin–plexin signaling pathway, a pathway not previously associated with SS that regulates both motility and differentiation of the immune and nervous systems. In support of this increase, we found increases in transcripts for *Sema4c* (FC = 4.12, *p* = 3.72 × 10^−4^), *Sema4d* (FC = 2.35, *p* = 2.03 × 10^−4^), *Sema6d* (FC = 3.56, *p* = 9.57 × 10^−4^), and their receptors *Plxnb1* (FC = 4.78, *p* = 2.33 × 10^−6^) and *PlxnD1* (FC = 3.12, *p* = 2.72 × 10^−4^; Table 1). At 5 wk, *Sema7a* transcripts (FC = 3.02, *p* = 0.00026, Table 1) were upregulated despite no change in total SEMA7A protein (5 wk: 1.16 ± 0.08 vs. 1.25 ± 0.18, *p* = 0.68). However, by 7 wk, Western blot showed upregulation of glycosylated (100 kDa), nonglycoslyated (75 kDa), and cleaved forms (50 kDa) during disease progression (7 wk: 1.09 ± 0.05 vs. 2.22 ± 0.19, *p* = 0.02, Figure 5A). As this pathway serves roles in both immune cell trafficking and axon guidance, we questioned whether innervation was different in the 5 wk Aire-/- compared to WT and 7 wk Aire-/-. As shown in Figure 5B, the extent of β3-tubulin (TUBB3)+ nerve fibers innervating the 5 wk tissue was reduced compared to WT LG and was similar to the 7 wk (WT 0.07 ± 0.01 vs. 5 wk 0.03 ± 0.004, *p* = 0.03, 7 wk: 0.02 ± 0/008, *p* = 0.01, Figure 5B). Despite clear depletion of axons in the Aire-/- LG by 5 wk, we found no significant reduction in acetylcholine production (Figure 5C), as measured via acetylcholinesterase assay (5 wk: 0.0098 ± 0.002 vs. 0.0068 ± 0.008, *p* = 0.20), suggesting that the remaining nerves showed a compensatory increase in function at the early disease stages to maintain tear secretion. However, with disease progression and further destruction of LG tissue, acetylcholine production was significantly reduced in 7 wk Aire-/- mice (7 wk: 0.019 ± 0.003 vs. 0.005 ± 0.001, *p* = 0.01, Figure 5C).

## 3. Discussion

Here, we reveal that in the earliest stages of autoimmune exocrinopathy, before functional output is compromised and corneal pathologies are visible, the lacrimal gland undergoes multiple phenotypic and molecular alterations associated with late-stage disease. These include tissue remodeling, such as the progressive loss of acinar cells, expansion of the ductal system, increased vascularization, and reduced innervation of the secretory epithelium. Phenotypic changes were accompanied by the enrichment of inflammatory mediators, such as IFNγ, IL1β, IL6, NF-κB, toll-like receptor, and JAK/STAT signaling pathways; as well as positive regulation of cell proliferation, lymphocyte differentiation, B and T cell activation, and pro-inflammatory macrophage expansion. In addition to confirming the presence of signaling pathways previously associated with late-stage disease in SS, we also revealed the presence of a novel signaling pathway, the semaphorin/plexin pathway, that is upregulated in the early stages of disease and may contribute to disease progression. Taken together, our data suggest that signaling pathways observed in the late stages of SS exocrinopathy are activated early during disease development and include a combination of pathways regulating inflammation, innervation, and cell survival.

Loss of corneal integrity and corneal inflammation as a result of reduced tear production is one of the major hallmarks of patients with SS [21]. Yet, whether loss of epithelial barrier function and increased corneal inflammation are direct readouts of lacrimal gland damage and dysfunction remains unclear. We found T and B cell infiltration and activation, pro-inflammatory macrophage expansion, and activation of multiple inflammatory pathways during disease development when only mild changes in the corneal epithelium were apparent (as noted by limited lissamine green staining) and before the acquisition of corneal inflammation or reduced tear secretion. Such an outcome has important implications for the early detection and diagnosis of dry eye disease. Indeed, our data suggest that diagnostic tests for aqueous-deficient dry eye are unlikely to identify patients until well after inflammatory events have been initiated, further increasing the need for early biomarkers of disease development.

Peripheral innervation is known to be essential to salivary gland homeostasis through the maintenance and activation of stem cells [22]. Although this specific role has not been evaluated in the LG, similar to the salivary gland, previous studies have shown that atrophy of LG acini occurs in the absence of a nerve supply [23]. We previously demonstrated that peripheral innervation is profoundly reduced in the Aire-/- cornea and lacrimal gland at 8 wk [24], an outcome that is consistent with reduced innervation observed in the human cornea [25,26]. Intriguingly, in the current study we unexpectedly found that acetylcholine synthesis was similar between WT and Aire-/- LG at 5 wk even though the number of nerve fibers within the Aire-/- LG at this age was reduced. This finding suggests that innervating nerves have compensatory function under conditions of mild inflammation that preserve tissue architecture as well as tear production. However, as the tissue becomes chronically inflamed, innervation is lost, resulting in organ dysfunction and tissue degeneration. Although the specific mechanisms behind the reduction in innervation are not known, one likely player is the destruction of secretory tissue. Epithelial cells synthesize neurotrophic factors such as GDNF and neurturin that serve to promote and maintain the nerve supply [27,28,29]. Although we did not find these molecules to be altered transcriptionally, whether these proteins are functional within highly inflamed tissue remains to be determined.

Sjögren’s Syndrome has long been considered a Th1-mediated disease elicited through the transformation of naïve CD4+ T cells into Th1 lymphocytes. Using the Aire-deficient mouse model, we have previously shown that IFNγ-secreting CD4 T cells cooperate with local IL1/IL1-receptor signaling pathways to induce aqueous tear deficiency and ocular surface disease [10,30]. In support of a Th1 response, our transcriptomic data in the 5 wk Aire-/- LG also identified the enrichment of Th1 cytokine signaling pathways and factors including IFNγ and TNFα. In addition, we found a dramatic upregulation of STAT1, a potent promoter of Th1 differentiation and maintainer of Th1 cytokine production [31], which has previously been shown to be overexpressed in salivary glands of SS patients [32]. Th1 activation and cytokine production occurred in conjunction with increased expression of CD68, suggesting that exposure of naïve monocytes to IFNγ or TNFα induces M1 (pro-inflammatory) development [33]. Notably, we found that these macrophages also express IL6R and that tissue showed increased production of its ligand, IL-6, both of which have previously been correlated with the histological grade of minor salivary gland biopsy [34]. Furthermore, blocking of IL6R has been shown to ameliorate autoimmune disease in a number of model systems (reviewed in [35]), suggesting that macrophage expansion may be a key driver of early disease. In support of this argument, our previous study demonstrated that ablation of macrophages in the Aire-/- led to significant improvements in lacrimal gland exocrinopathy and tear secretion [36]. Thus, early targeting of these cells may aid in reducing the pathogenesis of the condition.

Our data reveals the upregulation of a novel pathway not previously identified in murine models of SS—the semaphorin–plexin signaling pathway. Semaphorins have traditionally been associated with the nervous system, being active in the guidance of axons into tissues where they serve as either repulsive or attractive cues [37]. More recently, numerous semaphorins have been shown to function in the immune system, gaining the label “neuroimmune regulators”. One key role is serving as costimulatory molecules for T cell activation, an action that requires the presence of two signals: the first being TCR/MHC engagement, and the second being B7/CD28 interaction. With proper co-stimulation, three canonical signaling pathways—NF-κB, AP-1, and NFAT—are activated within T cells. These pathways lead to the expression of many molecules, such as chemokines, cytokines, and other cell surface molecules, that promote T cell activation. As such, they serve as a “signal three” for immune cell activation. In our data we identified significant upregulation of a number of semaphorins that are either constitutively expressed by T cells such as *Sema4d* and 6d (FC = 2.3 and 3.56, respectively, Table 1) or induced upon activation, namely SEMA7A (Table 1 and Figure 5). Upregulation of cell surface and serum levels of SEMA4D has also been measured in the MRL/lpr mouse model of SS, an outcome exhibiting a significant correlation with SS autoantibodies. These results suggest their value as a potential screening tool to monitor disease progression and/or assess the impact of a therapeutic intervention [16]. Moreover, given their known role in regulating the overall intensity of immune responses through the “signal three” activation, we predict that these molecules may be potential therapeutic targets for impairing disease progression.

In summary, our study defines disease progression at both the molecular and phenotypic level and highlights the need to identify early markers of disease progression. Indeed, our study suggests that functional outcomes currently being used to confirm disease status, including Schirmer’s test and lissamine green staining, do not faithfully predict the level of disease apparent within the lacrimal glands. It is likely that routine testing of tears for inflammatory molecules, such as SEMA4D, STAT1, TNFα, and IFNγ, may prove to be a more valuable assessment for identifying early disease onset. Moreover, the extent to which patient discomfort parallels changes in LG pathogenesis remains to be discerned.

## 4. Materials and Methods

### 4.1. Animal Model

Mice were handled in strict accordance with the University of California, San Francisco animal welfare guidelines for laboratory and animal care. The protocol was approved on 3 May 2018 by the Institutional Animal Care and Use Committee at the University of California San Francisco (Approval number: AN174695-01A). *Aire*-deficient mice on the BALB/c background (BALB/c Aire-/-) were the gift of Mark Anderson, University of California, San Francisco. Genomic DNA isolated from tail clippings was genotyped for the *Aire* mutations by PCR with manufacturer-recommended specific primers and their optimized PCR protocols. Female mice were used in the study and were between 5 and 7 weeks of age when sacrificed.

### 4.2. Immunohistological and Immunofluorescence Analysis

To visualize immune cell subtypes, immunohistochemistry was performed with antibodies specific for CD4 (BD Pharmigen, San Jose, CA, USA) and CD45R (eBiosciences, Waltham, MA, USA), a donkey anti-rat secondary antibody conjugated to horseradish peroxidase and a DAB (3,3′-Diaminobenzidine) staining kit. OCT Tissue Tek embedded lacrimal glands and eyes were sectioned on the cryostat (Leica, Izar, Germany) at 7 μm and mounted on SuperFront Plus slides. To fix the tissue, sections were incubated with ice-cold acetone at −20 °C for 10min and then washed for 5 min with PBS 3 times. Sections were blocked with 5% normal goat serum for 1hr at RT, and then incubated in 3% hydrogen peroxide to inactivate endogenous peroxidases. Primary CD4 (1:50) or CD45R (1:200) antibodies were prepared in 5% normal goat serum and incubated at 4 °C overnight. Sections were washed for 5 min with PBS 3 times, secondary antibody diluted 1:500 in 5% normal goat serum was applied for 30 min at room temperature (RT), and then sections were washed again for 5 min with PBS 3 times. Hematoxylin counterstain was utilized to stain the nuclei. For immunofluorescence staining, OCT Tissue Tek 7 or 20 μm sections were mounted and subsequently fixed with 4% paraformaldehyde for 20 min at RT, followed by 1 min acetone–methanol, and incubated overnight at 4 °C with one or a combinaton of the following primary antibodies: goat anti-COLIV (1:500, Millipore, Burlington, MA, USA), rat anti-PECAM (1:300 Millipore), goat anti-NKCC1 (1:300, Santa Cruz Biotech, Dallas, TX, USA), rabbit anti-cKit (1:200, Cell Signaling, Danvers, MA, USA), rabbit anti-MIST1 (1:1000, gift from Steven Konieczny), rabbit anti-cleaved Caspase 3 (1:200; Cell Signaling), rat anti-Ki67 (1:100, Dako, Santa Clara, CA, USA), rabbit anti-TUBB3 (1:500, Covance, Princeton, NJ, USA), rat anti-Ecadherin (1:300, Life Technologies, Carlsbad, CA, USA), rabbit anti-pStat3 (1:100, Cell Signaling), rat anti-IL6Ra(1:100, Thermo Fisher), Rat anti-CD68 (1:400, Bio-Rad, Hercules, CA, USA). Antibodies were detected using Cy2-, Cy3-, or Cy5-conjugated secondary Fab fragment antibodies (Jackson Laboratories, Sacramento, CA, USA), and nuclei were stained with Hoechst. Fluorescence was analyzed using a Zen Spinning disk confocal (Jena, Germany), Zeiss Epifluorescence microscope (Thornwood, NY, USA), and NIH ImageJ software (Bethesda, MD, USA).

CD4+ T cells were quantified in the central corneal region by counting the total number of DAB+ brown cells with an Hemotoxylin & Eosin counterstain associated nucleus. To quantify CD4+ T cells and CD45R+ B cells in the lacrimal gland, the area of DAB+ signal was measured and divided by the total area of the lacrimal gland. This value was expressed as a percentage of CD4+ T cell or CD45R+ B cell infiltration in the tissue.

In the lacrimal gland, MIST1 was utilized to label the acinar population while NKCC1 was utilized to label the ductal population. Lacrimal glands were imaged by confocal microscopy at a magnification of 20× with a stack of 20 1 μm slices. To determine the acinar to ductal cell ratio, the total number of MIST1+ cells associated with Hoeschst-stained nuclei was divided by the total number of NKCC1+ cells associated with Hoeschst-stained nuclei.

Blood vessels were labeled for PECAM and imaged by confocal microscopy at 20× using 1 μm confocal sections that were combined to produce a 20 μm projection. For assessing blood vessel dilation, blood vessel area was approximated with an ellipse. The blood vessel diameter was measured as the length of the approximated width for an approximated ellipse, expressed in micrometers. Blood vessels with a diameter greater than 6 μm were labeled as dilated. Blood vessel diameter in three 200 μm × 200 μm Ecadherin (ECAD)-positive areas of the tissue was quantified to determine total number of blood vessels 1–5 μm in diameter and >6 μm in diameter in intact gland. 

Twenty-micron lacrimal gland sections were labeled for beta3 tubulin (TUBB3) and ECAD, and imaged by confocal microscopy using a 20× objective with 1 μm confocal sections that were combined to give a projected thickness of 20 μm. Nerve density was measured as the area of TUBB3 immunofluorescent signal, as quantified using Tsai’s thresholding method (Moments) in ImageJ [38]. Integrated density was recorded with ImageJ. Total TUBB3 signal was normalized by the area of intact epithelial tissue, labeled using Ecadherin, and expressed as normalized nerve density. Total IL6Ra expression level was quantified using Tsai’s thresholding method, then normalized by the area of the tissue.

Ductal cell proliferation was quantified by counting the number of Ki67+ cells in the NKCC1+ ductal cells. To obtain the percentage of proliferating cells per respective region, total cell counts were divided by the number of NKCC1+ Hoechst-stained nuclei. Cell death was labeled using an antibody directed against Cleaved Caspase 3 (CC3). Apoptosis was calculated by counting the number of CC3+ cells in the NKCC1+ ducts and dividing by the total number of NKCC1+ ductal cells to obtain the percentage of apoptotic ductal cells. To obtain the percentage of acinar cell death, the number of CC3+ ECAD+ NKCC1- acinar cells was counted and divided by the total number of ECAD+NKCC1- cells. The result was used to quantitatively assess the loss of acinar cells and the expansion of the ductal cell compartment. Lastly, the percentage of pSTAT3+ ductal cells was obtained by counting the number of pSTAT3+ NKCC1+ cells and dividing by the total number of NKCC1+ ductal cells.

### 4.3. Western Blotting and Analysis

Tissues were lysed in radioimmunoprecipitation assay buffer (RIPA buffer) containing a cocktail of protease inhibitors. Protein concentration was measured by bicinchoninic acid (BCA) assay (Thermo Scientific, Waltham, MA, USA). A quantity of 10 μg of protein per samples was separated by sodium dodecyl sulfate polyacrylamide gel electrophoresis (SDS-PAGE) on 4–12% Bis Tris gels (Invitrogen, Carlsbad, CA, USA) and transferred to PVDF membranes. Membranes were blocked in 5% milk and incubated in primary antibodies overnight at 4 °C: rabbit anti-Sema7a (1:500, Abcam, Cambridge, MA, USA), Rabbit anti-Stat1 (1:1000, Cell Signaling), and mouse anti-Beta actin (1:1000, Sigma, St. Louis, MO, USA). After washing three times in TBST, the blots were incubated with appropriate horseradish-peroxidase-conjugated secondary antibodies and visualized with the Clarity Max Western ECL chemiluminescence detection system (Biorad, Hercules, CA, USA). Densitometric quantification of bands was performed using ImageJ software. To correct for sample-to-sample variability, the density of the target protein band was normalized to the density of the loading control beta actin band, for comparison of relative protein levels.

### 4.4. Lissamine Green Staining of the Ocular Surface

Five microliters of lissamine green dye (1%)were applied to the lower conjunctival cul-de-sac of mice anesthetized with isoflurane, and images of the cornea were taken using an Olympus Zoom Stereo Microscope (Olympus, Center Valley, PA, USA). Corneas were divided into four quadrants and scored independently for the extent of staining in each quadrant. Scores were classified as Grade 0, no staining; Grade 1, sporadic (<25%); Grade 2, diffuse punctate (25–75%), or Grade 3, coalesced punctate staining (75% or more). The sum of the quadrants scored by three masked observers for each eye was calculated on a scale ranging from 0 (no staining) to 12 (most severe staining) and plotted as the fold change relative to the average WT score from the same age group.

### 4.5. Tear Secretion Measurement

Pilocarpine (4.5 mg/kg) was injected into the peritoneum (i.p), and after ten minutes, the mice were anesthetized with isoflurane. Tear secretion was measured using a Zone-Quick phenol red thread (Showa Yakuhin Kako Co. Ltd., Tokyo, Japan), to indicate mm of tears absorbed in 15 s.

### 4.6. RNA Isolation and RNAseq Analysis

Total RNA was collected and purified using RNAaqueous and DNase reagents according to the manufacturer’s instructions (Ambion, Houston, TX, USA). For each RNA sample (RNA Integrity Number (RIN) >6), cDNA libraries were prepared using the TruSeq mRNA library prep kit (Illumina Inc, San Diego, CA, USA) using 1 µg RNA according to the manufacturer’s instructions and were then 50 bp single-end sequenced on an Illumina HiSeq 4000. Quality control metrics were performed on raw sequencing reads using the FASTQC v0.11.6 application [39]. Reads were mapped to the Mus musculus genome (mm10 build) using Spliced Transcripts Alignment to a Reference (STAR) [17]. At least 90% of the reads were successfully mapped. Reads aligning to the University of California, Santa Cruz (UCSC) mm10 build were quantified against Ensembl Transcripts release 93 using Partek^®^ E/M (Partek’s optimization of the expectation maximization algorithm, Partek Inc, St.Louis, MO, USA), which disregarded any reads that aligned to more than one location or more than one gene at a single location. Data was normalized by two procedures: 1. total count normalization, 2. addition of a small offset (0.0001). After normalizing, differential gene expression was determined using the Differential gene expression (GSA) algorithm (Partek). This algorithm generates *p* values using limma-trend method 4. Genes were considered differentially expressed if the log_2_ Fold Change between samples was at least 1, with the adjusted *p*-value held to 0.01. Additionally, we considered only genes with values of at least ten fragments per kilobase of transcript per million mapped reads (FPKM) in at least one biological replicate. A gene list of all significantly modulated genes (*p* < 0.01) was used as input for gene ontology (GO) analysis using Partek software. The resulting output was ranked by adjusted *p*-value (Fisher’s exact test on the underlying contingency table). We consider an attribute to be significant if its Expression Analysis Systematic Explorer (EASE) score (Fisher exact test *p* value) is less than 0.05 relative to an appropriate background gene set. Gene sets were selected to highlight unique features of the enrichment analysis. A principal component analysis (PCA) was performed on normalized log_2_-transformed read counts using seven principal components. The dataset is available on the GEO database.

### 4.7. Acetylcholinesterase Assay

To assess acinar cell function, the Amplex Red Acetylcholine/Acetylcholinesterase Assay Kit (A12217) (Molecular Probes, Eugene, OR, USA), was used to measure acetylcholinesterase production in three 20 μm fresh frozen OCT sections of the lacrimal gland.

### 4.8. Statistical Analysis 

A minimum of three independent repeats were conducted in all experiments. Bar graphs are used to summarize the means and standard errors of each outcome obtained using all data collected from WT and *Aire*-/- mice. Data are presented as mean ± SEM. Student’s *t*-test was used for two-group comparisons. One-way ANOVA was used for comparison across multiple groups; *p* ≤ 0.05 was considered statistically significant.

## Figures and Tables

**Figure 1 ijms-19-03628-f001:**
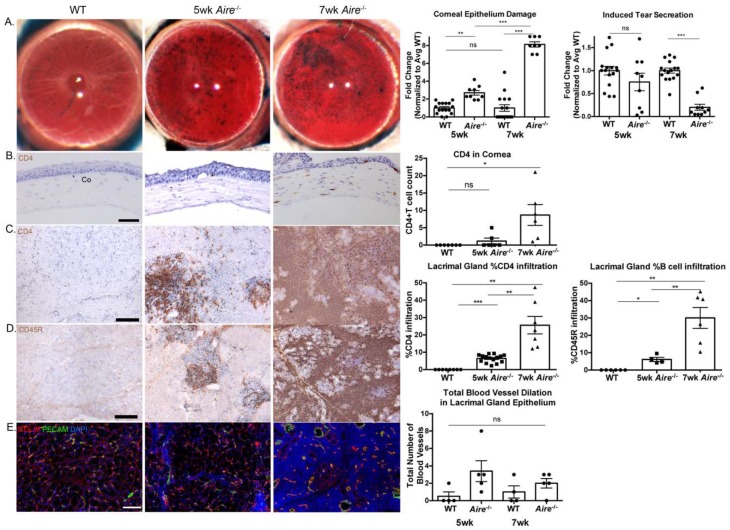
Disease progression in the Aire-/- mouse as shown by (**A**) increased lissamine green staining of the cornea and reduced tear secretion, (**B**) extensive CD4+ T cell infiltration of the cornea, (**C**) expansion of CD4+ T cell- and (**D**) CD45R+ B cell-containing foci in the lacrimal gland, and (**E**) altered tissue structure and blood vessel dilation in the lacrimal gland indicated by collagen type IV/ platelet and endothelial cell adhesion molecule (COLIV/PECAM) staining. Data are expressed as mean ± SEM. *n* = minimum of 4 mice per group, and each sample is represented by a circle, square, or triangle within a group. Changes in Lissamine green and tear secretion are expressed as fold change relative to average WT. Controls in (**B**–**D**) included 5-week-old (wk) and 7 wk WT as the WT at both ages did not show any CD4+ T cell or CD45R+ B cell infiltration. * *p* < 0.05, ** *p* < 0.01, *** *p* < 0.001. Scale bar = 100 µm.

**Figure 2 ijms-19-03628-f002:**
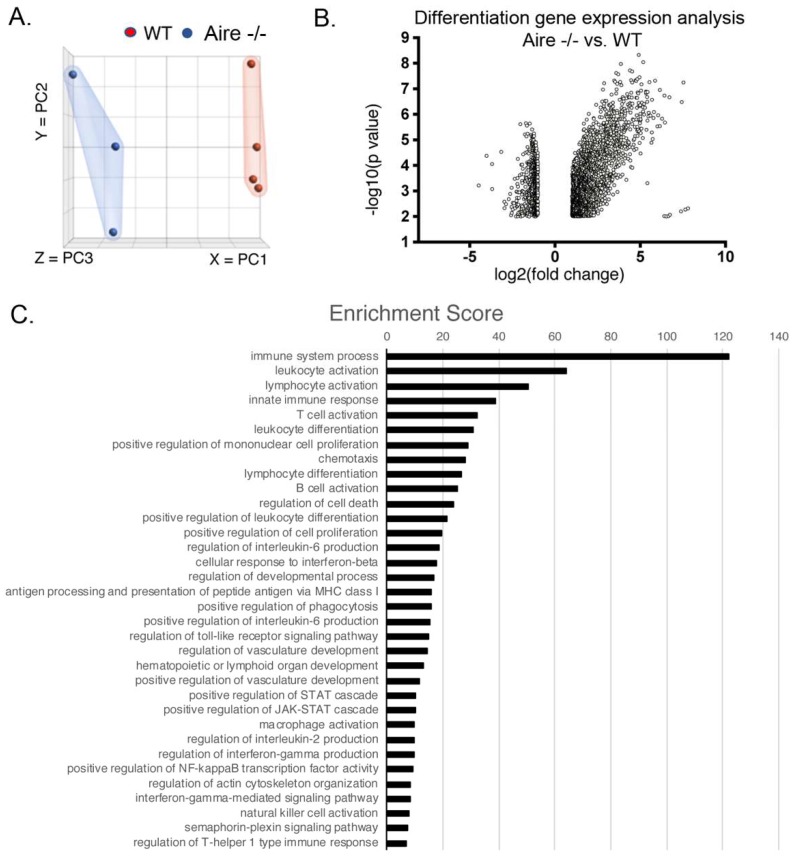
RNA sequencing (RNAseq) of the 5 wk Aire-/- lacrimal glands revealed genes significantly altered during early disease, implying that these genes could serve as early disease markers. (**A**) The Principal Component Analysis (PCA) plot shows clusters of samples. (**B**) The volcano plot shows number and magnitude of genes significantly up- and down-regulated (above and below; FC = 2, respectively) during early disease. (**C**) Ranked list of biological pathways enriched during the early stages of disease.

**Figure 3 ijms-19-03628-f003:**
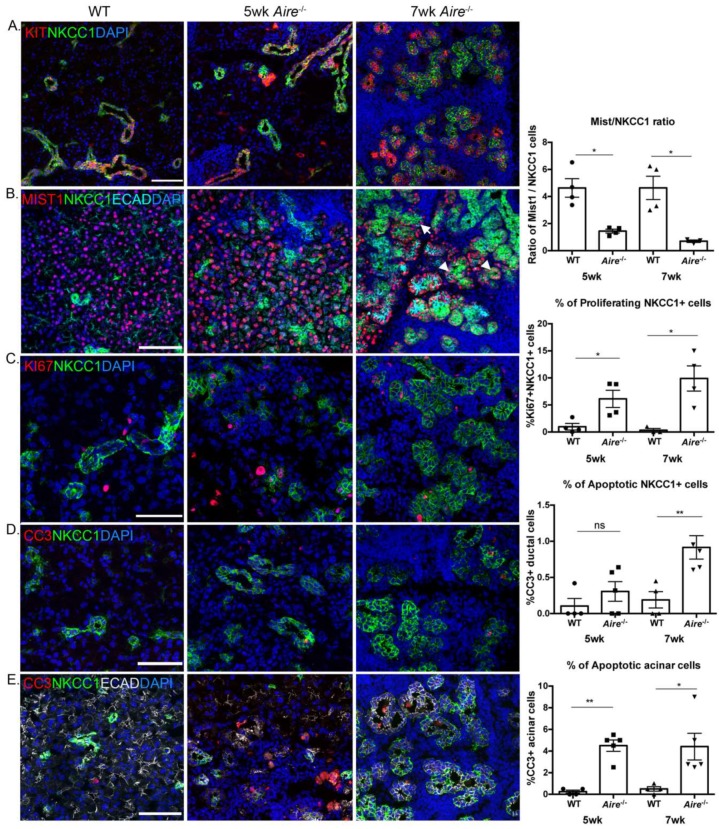
Structural changes in the Aire-/- lacrimal gland. (**A**) Increased presence of ductal cell markers NKCC1 and KIT expression. (**B**) Gradual loss of Mist1+ acinar cells with expansion of NKCC1+ cells. Arrowheads indicate MIST1+ acinar cells co-expressing ductal marker NKCC1. (**C**) Increased ductal cell proliferation and (**D–E**) increased apoptotic acinar cells in Aire-/- lacrimal gland. Data are expressed as mean ± SEM. Data in **A** are expressed as ratio of MIST- to NKCC1-positive cells. Data in (**B–E**) are expressed as percentages of proliferating or apoptotic cells. *n* = 3–4 mice per group, and each sample is represented by a circle, square, or triangle within a group. * *p* < 0.05, ** *p* < 0.01. Scale bar = 100 µm.

**Figure 4 ijms-19-03628-f004:**
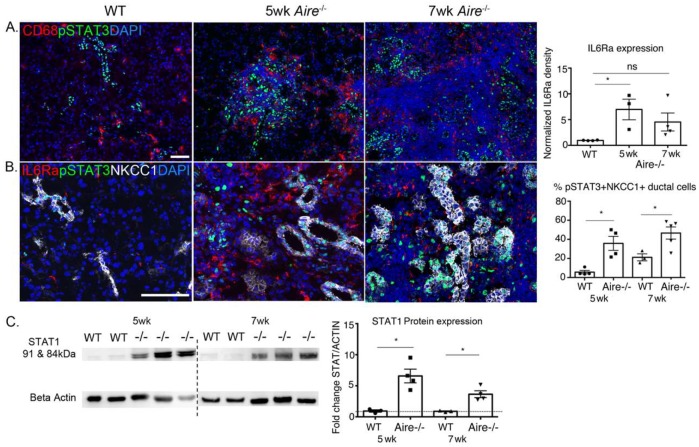
STAT signaling increased with disease progression in Aire-/- mice. (**A**) pSTAT3 is increased in CD68+ inflammatory macrophages and (**B**) NCKK1+ ductal cells of the inflamed LG by 5 wk and increases with disease progression. (**B**) IL6Ra expression is increased in infiltrating immune cells. (**C**) Increased levels of STAT1 protein in the Aire-/- LG were confirmed by Western blot. Data are expressed as mean ± SEM. *n* = 3–4 mice per group, and each sample is represented by a circle, square, or triangle within a group. * *p* < 0.05. Scale bar = 100 µm.

**Figure 5 ijms-19-03628-f005:**
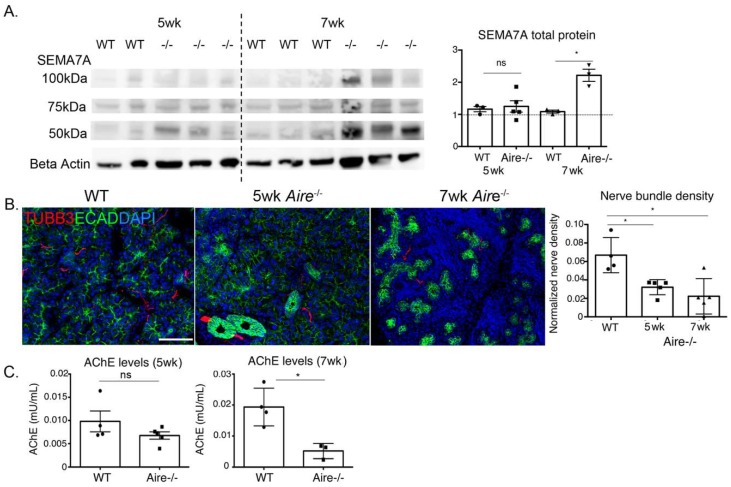
Lacrimal gland innervation and activation of semaphorin/plexin signaling was altered in Aire-/- mice. (**A**) Western blot showed no change in axon guidance factor semaphorin 7A (SEMA7A) at 5 wk but revealed upregulation of SEMA7A and increased levels of cleaved SEMA7A at 50 kDa at 7 wk. (**B**) A significant reduction in LG innervation was observed during early disease. (**C**) However, nerve function measured by acetylcholine esterase (AChE) assay was not altered until late disease. Data in **A** are expressed as fold change in protein expression relative to WT. Data in **B**,**C** are expressed as mean ± SEM. *n* = 3–7 mice per group and each sample is represented by a circle, square, or triangle within a group. * *p* < 0.05. Scale bar = 100 µm.

**Table 1 ijms-19-03628-t001:** List of representative genes that are significantly altered in the Aire-/- lacrimal gland during early disease onset from RNAseq showing fold change and *p*-value compared to wild type.

Cell/Process	Gene Symbol	Gene Description	Fold Change (-/- vs. WT)	*p*-value (-/- vs. WT)
Immune cell				
Adaptive	*Cd79b*	B-cell antigen receptor complex-associated protein β chain	200.00	5.35 × 10^−3^
	*Cd19*	B-lymphocyte antigen CD19	172.92	3.30 × 10^−7^
	*Cd3d*	T-cell surface glycoprotein CD3 δ chain	106.92	8.57 × 10^−3^
	*Cd3g*	T-cell surface glycoprotein CD3, gamma polypeptide	100.81	9.69 × 10^−3^
	*Lat*	linker for activation of T cells	42.93	1.76 × 10^−6^
Innate	*Srgn*	Serglycin (mast cell)	5.70	4.65 × 10^−6^
	*Ncf1*	neutrophil cytosolic factor 1	12.36	1.67 × 10^−5^
	*Klrk1*	killer cell lectin-like receptor subfamily K, member 1	32.18	1.23 × 10^−6^
	*Cd68*	CD68 antigen	3.65	1.95 × 10^−5^
Vascular	*Vegfd*	vascular endothelial growth factor D	3.70	1.03 × 10^−3^
	*Pecam1*	platelet/endothelial cell adhesion molecule 1	2.67	4.3 × 10^-4^
	*Vcam1*	vascular cell adhesion molecule 1	23.10	5.68 × 10^−8^
Proliferation	*Aurkb*	aurora kinase B	20.12	1.26 × 10^−5^
	*Top2a*	topoisomerase (DNA) II α	3.69	3.17 × 10^−4^
	*Tpx2*	Microtubule Nucleation Factor, microtubule-associated	11.59	2.28 × 10^−5^
Cell death	*Bid*	BH3 interacting domain death agonist	9.69	6.13 × 10^−6^
	*Casp3*	caspase 3	3.73	7.69 × 10^−5^
	*Trp53*	transformation related protein 53	2.30	2.60 × 10^−3^
Cell migration	*Cxcl9*	chemokine (C–X–C motif) ligand 9	167.93	6.30 × 10^−3^
	*Cxcr3*	chemokine (C–X–C motif) receptor 3	43.23	1.34 × 10^−7^
	*Csf1r*	colony stimulating factor 1 receptor	3.25	1.30 × 10^−4^
	*Sell*	selectin, lymphocyte	223.19	4.79 × 10^−3^
Axon guidance/ immunoregulation	*Sema4a*	sema domain, immunoglobulin domain (Ig), transmembrane domain (TM) and short cytoplasmic domain, (semaphorin) 4A(Sema4a)	2.42	9.17 × 10^−4^
	*Sema4c*	sema domain, immunoglobulin domain (Ig), transmembrane domain (TM) and short cytoplasmic domain, (semaphorin) 4C(Sema4c)	4.12	3.72 × 10^−4^
	*Sema4d*	sema domain, immunoglobulin domain (Ig), transmembrane domain (TM) and short cytoplasmic domain, (semaphorin) 4D(Sema4d)	2.35	2.03 × 10^−4^
	*Sema6d*	sema domain, transmembrane domain (TM), and cytoplasmic domain, (semaphorin) 6D	3.56	9.57 × 10^−4^
	*Sema7a*	sema domain, immunoglobulin domain (Ig), and GPI membrane anchor, (semaphorin) 7A	3.02	2.62 × 10^−4^
	*Plxnb1*	plexin B1	4.78	2.33 × 10^−6^
	*Plxnd1*	plexin D1	3.12	2.72 × 10^−4^
Secretion	*Scgb1b19*	secretoglobin, family 1B, member 19	−4.61	3.33 × 10^−3^
	*Pnliprp1*	pancreatic lipase related protein 1	−4.08	8.44 × 10^−3^
	*Scgb1b3*	secretoglobin, family 1B, member 3	−3.57	2.24 × 10^−4^
	*Scgb1b20*	secretoglobin, family 1B, member 20	−3.09	7.62 × 10^−5^
	*Pip*	prolactin induced protein	−2.31	3.22 × 10^−4^
	*Bhlha15*	basic helix–loop–helix family, member a15(Bhlha15)	−2.04	7.58 × 10^−3^
	*Esp6*	exocrine gland secreted peptide 6	−2.39	5.81 × 10^−3^
Signaling pathways				
TNF	*Tnfrsf18*	tumor necrosis factor receptor superfamily, member 18	33.63	3.64 × 10^−6^
	*Tnfrsf13b*	tumor necrosis factor receptor superfamily, member 13b	27.71	1.09 × 10^−6^
NF-κB	*Nfkbie*	nuclear factor of kappa light polypeptide gene enhancer in B cells inhibitor, epsilon	28.23	4.08 × 10^−6^
	*Nfkbiz*	nuclear factor of kappa light polypeptide gene enhancer in B cells inhibitor, zeta	2.90	1.29 × 10^−4^
	*Eomes*	eomesodermin	16.56	4.90 × 10^−6^
	*Pim1*	proviral integration site 1	6.25	1.17 × 10^−5^
Toll-like	*Tlr2*	toll-like receptor 2	10.24	3.30 × 10^−6^
	*Tlr6*	toll-like receptor 6	5.63	1.17 × 10^−3^
	*Tlr7*	toll-like receptor 7	14.87	5.74 × 10^−6^
	*Tlr13*	toll-like receptor 13	13.19	2.39 × 10^−4^
JAK/STAT	*Jak2*	Janus kinase 2	4.31	2.75 × 10^−5^
	*Jak3*	Janus kinase 3	25.17	9.18 × 10^−6^
	*Stat1*	signal transducer and activator of transcription 1	12.07	2.42 × 10^−8^
	*Stat2*	signal transducer and activator of transcription 2	3.65	1.92 × 10^−5^
Interleukin	*Il1b*	interleukin 1β	12.19	3.16 × 10^−5^
	*Il2ra*	interleukin 2 receptor, α chain	9.35	2.49 × 10^−4^
	*Il2rg*	interleukin 2 receptor, γ chain	21.03	3.65 × 10^−7^
	*Il6ra*	interleukin 6 receptor, α	7.40	9.40 × 10^−5^
Interferon	*Igtp*	interferon gamma induced GTPase	23.07	2.11 ×10^−8^
	*Irf1*	interferon regulatory factor 1	8.23	2.06 × 10^−6^
	*Irf5*	interferon regulatory factor 5	11.99	4.43 × 10^−6^

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
