# Peer review of "Deciphering Molecular and Phenotypic Changes Associated with Early Autoimmune Disease in the Aire-Deficient Mouse Model of Sjögren’s Syndrome"

_ijms, 2018, doi:10.3390/ijms19113628_

Round 1
Reviewer 1 Report
In the Aire-deficient mouse, the authors identify 5 weeks of age as the early stage of SSc-like disease and 7 weeks of age wherein functional deficiencies in tear secretion, reduced barrier function and lymphocyte infiltration are significantly changed. RNAseq was performed on lacrimal glands at 5 weeks in KO and WT animals. Differences in STAT signalling and the semaphorin-plexin signalling pathway were subsequently confirmed by immunoblotting and IHC.
Although the validation experiments did not show a great deal of difference between week 5 and 7, this is an important first step towards developing early interventions for SSc.
~Minor points~
The authors could improve Fig. 1 by expanding the legend to provide more explicit detail regarding part B-D, eg B is cornea, C,D&E are lacrimal gland, B&C are CD4 staining, D is CD45R and E is COLIV/PECAM/DAPI.
Page 4 Line 139
Use Greek symbols gamma and kappa in IFNγ and NFκB.
Page 11 Line 284
Change gamma to γ to be consistent.
Page 13 Line 347
In blocking ?buffer? overnight...
Page 13 Line
imageJ -> ImageJ
Is Appendix A Table1A essentially the same as Figure 2C? Could one of these be omitted?
The authors state in the Statistical Analysis section that a P value less than or equal to 0.05 was considered significant, but the P<0.05 is the accepted standard.
~Major points~
Page 14 Line 433
The authors' explanation of their RNAseq analysis could be expanded upon. As it is written, the authors have used the Partek software to analyse the data, but it is unclear which method has been used to determine the adjusted p value. Is this Benjamini & Hochberg or Bonferroni etc? The GO analysis and PCA are also light on detail.
Figure 4 C
The protein loading in the immunoblot for beta actin does not look particularly even. The KO mice appear to have significantly more protein than WT mice at week 5 and 7. The authors should comment on this.
Author Response
~Minor points~
Point 1: The authors could improve Fig. 1 by expanding the legend to provide more explicit detail regarding part B-D, eg B is cornea, C,D&E are lacrimal gland, B&C are CD4 staining, D is CD45R and E is COLIV/PECAM/DAPI.
Response 1: Figure 1 legends was updated to meet the reviewer’s suggestions as follows:
Figure 1. Disease progression in the Aire-/- mouse as shown by (A) increased lissamine green staining of the cornea and reduced tear secretion,(B) extensive CD4+ T cell infiltration of the cornea, (C) expansion of CD4+ T cell- and (D) CD45R+ B cell-containing foci in the lacrimal gland, and (E) altered tissue structure and blood vessel dilation in the lacrimal gland indicated by COLIV/PECAM staining. Data are expressed as mean±SEM. n=minimum of 4 mice per group. Change in Lissamine green and tear secretion are expression as fold change relative to average WT. Controls in B-D included 5wk and 7wk WT as the WT at both ages did not show any CD4+ T cell or CD45R+ B cell infiltration. WT*p<0.05, **p<0.01, ***p<0.001. Scale bar=100 µm.
Point 2: Page 4 Line 139
Use Greek symbols gamma and kappa in IFNγ and NFκB.
Response 2:We have replaced gamma and kappa with Greek symbols γ and κ.
Point 3:Page 11 Line 284
Change gamma to γ to be consistent.
Response 3: We have replaced gamma with Greek symbol γ.
Point 4: Page 13 Line 347
In blocking ?buffer? overnight...
Response 4:We apologize for the missing information and have revised the text in the methods as follows:
To visualize immune cell subtypes, immunohistochemistry was performed with antibodies specific for CD4 (BD Pharmigen) and CD45R (eBiosciences), a donkey anti-rat secondary antibody conjugated to HRP, and a DAB staining kit. OCT Tissue Tek embedded lacrimal glands and eyes were sectioned on the cryostat (Leica, Izar, Germany) at 7mm and mounted on SuperFront Plus slides. To fix the tissue, sections were incubated with ice cold acetone at -20°C for 10min and then washed for 5min with PBS 3 times. Sections were blocked with 5% normal goat serum for 1hr at RT, and then incubated in 3% hydrogen peroxide to inactivate endogenous peroxidases. Primary CD4 (1:50) or CD45R (1:200) antibodies were prepared in 5% normal goat serum and incubated at 4°C overnight. Sections were washed for 5min with PBS 3 times, secondary antibody diluted 1:500 in 5% normal goat serum was applied for 30min at RT, and then sections were washed again for 5min with PBS 3times. Hematoxylin counterstain was utilized to stain the nuclei. For immunofluorescence staining, OCT Tissue Tek 7mm or 20mm sections were mounted and subsequently fixed with 4% PFA for 20min at RT, followed by 1 min acetone-methanol, incubated overnight at 4°C with one or combination of the following primary antibodies: goat anti-COLIV (1:500, Millipore), rat anti-PECAM (1:300 Millipore), goat anti-NKCC1 (1:300, Santa Cruz Biotech), rabbit anti-cKit (1:200, Cell Signaling), rabbit anti-MIST1 (1:1000, gift from Steven Konieczny), rabbit anti-cleaved Caspase 3 (1:200; Cell Signaling), rat anti-Ki67 (1:100, Dako), rabbit anti-TUBB3 (1:500, Covance), rat anti-Ecadherin (1:300, Life Technologies), rabbit anti-pStat3 (1:100, Cell Signaling), rat anti-IL6Ra(1:100, Thermo Fisher), Rat anti-CD68 (1:400, Bio-Rad). Antibodies were detected using Cy2-, Cy3- or Cy5-conjugated secondary Fab fragment antibodies (Jackson Laboratories), and nuclei were stained with Hoechst. Fluorescence was analyzed using a Zen Spinning disk confocal, Zeiss Epifluorescence microscope, and NIH ImageJ software.
Point 5: Page 13 Line
imageJ -> ImageJ
Response 5:We have changed imageJ to ImageJ.
Point 6:Is Appendix A Table1A essentially the same as Figure 2C? Could one of these be omitted?
Response 6:Appendix A Table 1A provides p-values for each of the enriched biological pathways in addition to the enrichment score. We feel these data add substantively to the rigor of the manuscript and have chosen to include them as a separate table.
Point 7: The authors state in the Statistical Analysis section that a P value less than or equal to 0.05 was considered significant, but the P<0.05 is the accepted standard.
Response 7:P-values fall on a continuous scale and must be interpreted in the context of each specific study. Based on the sample size and sensitively of our analysis, we chose P£0.05 to determine significance. In reviewing the data, the only finding impacted by our choice was the comparison in expression level of total SEMA7A protein between 5wk WT vs. 5wk KO. We believe this pathway, whether significant or borderline significant is worthy of highlighting.
~Major points~
Point 8:Page 14 Line 433
The authors' explanation of their RNAseq analysis could be expanded upon. As it is written, the authors have used the Partek software to analyse the data, but it is unclear which method has been used to determine the adjusted p value. Is this Benjamini & Hochberg or Bonferroni etc? The GO analysis and PCA are also light on detail.
Response 8: We have added details to the Materials and Methods section as follows:
Total RNA was collected and purified using RNAaqueous and DNase reagents according to manufacturer’s instructions (Ambion, Houston, TX). For each RNA sample (RIN > 6), cDNA libraries were prepared using the TruSeq mRNA library prep kit (Illumina Inc) using 1µg RNA according to the manufacturers instructions and were then 50 bp single-end sequenced on an Illumina HiSeq 4000. Quality control metrics were performed on raw sequencing reads using the FASTQC v0.11.6 application. Reads were mapped to the Mus musculus genome (mm10 build) using Spliced Transcripts Alignment to a Reference (STAR).19At least 90% of the reads were successfully mapped. Reads aligning to the UCSC mm10 build were quantified against Ensembl Transcripts release 93 using Partek® E/M (Partek’s optimization of the expectation-maximization algorithm, Partek Inc), which disregarded any reads that aligned to more than one location, or more than one gene at a single location. Data was normalized by two procedures: 1. total count normalization, 2. addition of a small offset (0.0001). After normalizing differential gene expression was determined using the Differential gene expression (GSA) algorithm (Partek). This algorithm generates p values using limma-trend method 4. Genes were considered differentially expressed if the log2 Fold Change between samples was at least 1, with the adjusted p-value held to 0.01. Additionally, we considered only genes with values of at least ten FPKM in at least one biological replicate. A gene list of all significantly modulated genes (p<0.0< span="">1) were used as input for gene ontology (GO) analysis using Partek software. The resulting output was ranked by adjusted p-value (Fisher’s exact test on the underlying contingency table). We consider an attribute to be significant if its Expression Analysis Systematic Explorer (EASE) score (Fisher exact test P value) is less than 0.05 relative to an appropriate background gene set. Gene-sets were selected to highlight unique features of the enrichment analysis. A principalcomponent analysis (PCA) was performed on normalized log2 transformedread counts using 7 principle components. The dataset is available on the GEO database.
Point 9: Figure 4 C
The protein loading in the immunoblot for beta actin does not look particularly even. The KO mice appear to have significantly more protein than WT mice at week 5 and 7. The authors should comment on this.
Response 9: We agree with the reviewer on this point. To mathematically compensate and correct for this variation, when conducting the densitometric quantification using ImageJ, the density of the STAT1 band was normalized to the density of the beta actin band in each sample. This normalization method has been added to the Material and Methods under section 4.3 Western blotting and Analysis.

Reviewer 2 Report
Excellent study, clearly described, with novel findings, and of interest to researchers in this area.
Aire-/- mice are available on multiple different genetic backgrounds ( B6, NOD, and Balb/c). The authors have previously published characterization of LG and corneal disease using Aire-/- mice on NOD background.
It would be useful to include in the introduction or discussion section a comment on why BALB/c Aire-/- mice were selected for this study, and how the BALB/c mice differ from the Aire-/- on B6 or NOD backgrounds based on their disease characteristics.
Author Response
Point 1: Aire-/- mice are available on multiple different genetic backgrounds ( B6, NOD, and Balb/c). The authors have previously published characterization of LG and corneal disease using Aire-/- mice on NOD background.
It would be useful to include in the introduction or discussion section a comment on why BALB/c Aire-/- mice were selected for this study, and how the BALB/c mice differ from the Aire-/- on B6 or NOD backgrounds based on their disease characteristics.
Response 1: We thank the reviewer for the constructive comments and suggestions. The following text has been added to the Introduction, line 70: “ Aire -/- mice on the BalbC background were selected over Aire-/- mice on NOD background for this study due to their less rapid development of severe disease. NOD. Aire-/- develop multi-organ autoimmune disease that results in an average life span of ~8 weeks. In contrast, BalbC. Aire -/- mice develop SS-associated dry eye and exocrinopathy that is nearly identical in appearance to NOD.Aire-/-, but the time course of disease development is slower, the extent of multi-organ involvement is less, and the average life span extends up to ~16 weeks. These characteristics make BalbC. Aire-/- mice an ideal model for studies of disease progression.”